# The Practicability of the Xpert HCV Viral Load Fingerstick Point-of-Care Assay in Primary Care Settings

**DOI:** 10.3390/v13112327

**Published:** 2021-11-22

**Authors:** David Petroff, Olaf Bätz, Katrin Jedrysiak, Jan Kramer, Thomas Berg, Johannes Wiegand

**Affiliations:** 1Clinical Trial Centre, University of Leipzig, 04107 Leipzig, Germany; david.petroff@zks.uni-leipzig.de; 2LADR Laboratory Group Dr. Kramer & Colleagues, 21502 Geesthacht, Germany; o.baetz@ladr.de (O.B.); k.jedrysiak@ladr.de (K.J.); j.kramer@ladr.de (J.K.); 3Division of Hepatology, Department of Medicine II, Leipzig University Medical Center, 04103 Leipzig, Germany; thomas.berg@medizin.uni-leipzig.de

**Keywords:** hepatitis C, screening, elimination, World Health Organization, check-up

## Abstract

Linkage to care presents one obstacle toward eliminating HCV, and the current two-step pathway (anti-HCV, followed by HCV-RNA testing) results in the loss of patients. HCV screening was tested in the primary care setting with the fingerstick Xpert HCV viral load point-of-care assay to analyze the practicability of immediate diagnosis. Anti-HCV (Cobas) and HCV-RNA (Cobas Amplicor version 2.0, only performed if anti-HCV was positive) were analyzed centrally as the gold standard. The Xpert assay was performed by 10 primary care private practices. In total, 622 patients were recruited. Five individuals (0.8%) were anti-HCV positive, and one was HCV-RNA positive. The Xpert test was valid in 546/622 (87.8%) patients. It was negative in 544 and positive in 2 cases, both of whom were anti-HCV negative. The HCV-RNA PCR and the Xpert test were both negative in 4/5 anti-HCV-positive cases, and the individual with HCV-RNA 4.5 × 10^6^ IU/mL was not detected by the Xpert test. Primary care physicians rated the Xpert test practicability as bad, satisfactory, or good in 6%, 13%, and 81%, respectively, though 14/29 (48%) bad test ratings were assigned by a single practice. Despite adequate acceptance, interpretability and diagnostic performance in primary care settings should be further evaluated before its use in HCV screening can be recommended.

## 1. Introduction

Five years ago, the World Health Organization presented a strategy for eliminating chronic hepatitis C virus (HCV) infection by the year 2030 [1]. In Germany, this goal is supported by the Ministry of Health with the BIS2030 initiative [2]. Models show the importance of adequate treatment uptake with modern direct antiviral agents [3], which led to the wide use of interferon-free antiviral therapies in clinical practice: German registry data show sustained virological response rates of 97% and favorable tolerability of various treatment regimens [4]. These major achievements should be accompanied by screening and linkage-to-care programs [3]. However, current test algorithms are based on a two-step pathway with detection of anti-HCV to confirm exposure, followed by HCV-RNA to detect active infection in a subsequent visit, which leads to a drop-off in HCV-RNA-positive patients, estimated to comprise only 68% of these patients [5]. The HCV care continuum is further impaired, because HCV-RNA-positive patients are usually referred from primary care to the secondary care of hepatologists or infectious disease specialists: Follow-up data of an HCV screening project with primary care physicians showed that subsequent HCV specific clinical management is sub-optimal [6], and recent Italian primary care data indicate that 39% of the HCV population receive antiviral therapy, while only 19% are referred to specialists and 42% are not [7]. Therefore, point-of-care assays enabling detection of HCV-RNA and diagnosis of active infection in a single visit are important for clinical use.

The Xpert HCV-RNA viral load fingerstick assay is a CE-marked test developed to improve the diagnosis of viremic infection at the point of care without the need for a follow-up visit. Sensitivity and specificity for HCV-RNA detection in samples collected by fingerstick are at least 95.5% and 98.1% in comparison with the Abbott RealTime HCV-RNA PCR as a standard diagnostic tool [8,9]. The assay has been evaluated in high-risk populations in drug and alcohol clinics, homelessness services, needle and syringe exchange programs in people who inject drugs, and prisoners [8,9,10,11,12,13]. A WHO report notes that the assay “is intended for use as an aid in the management of HCV infected patients undergoing antiviral therapy” and “is not intended to be used as a blood donor screening test for HCV” (WHO Prequalification of In Vitro Diagnostics PUBLIC REPORT, Product: Xpert^®^ HCV Viral Load with GeneXpert^®^ Dx, GeneXpert^®^ Infinity-48s, and GeneXpert^®^ Infinity-80 WHO reference number: PQDx 0260-070-00). However, since it “quantifies HCV genotypes 1–6 over the range of 10 to 100,000,000 IU/mL” and recent European recommendations state that “screening for HCV infection can be replaced by low-cost point-of-care tests for viraemia with a lower limit of detection 1000 IU/mL”, it is important to generate evidence for its diagnostic properties in the screening setting. The aim of this study was thus to analyze its practicability on the primary care level, a setting that has not yet been explored.

## 2. Methods

The Xpert HCV Viral Load Fingerstick assay was performed in primary care private practices as follows: After fingerstick, 100 μL of capillary whole blood was immediately collected with a minivette and placed directly into the Xpert HCV VL FS assay cartridge (Cepheid, Krefeld, Germany; lower limit of detection 40 IU/mL, lower limit of quantification of 100 IU/mL) for on-site HCV-RNA testing. The cartridge was loaded into a GeneXpert Instrument. The time to result is about 60 min [8,9]. If the practices reported that the test did not provide a clear result, we referred to it as “invalid”.

The practicability of the test was evaluated with a questionnaire including a four-point Likert scale.

After venipuncture, alanine and aspartate aminotransferase (ALT, AST; upper limit of normal 50 U/L for males, 35 U/L for females), gamma-glutamyltransferase (GGT; upper limit of normal 60 U/L for males, 40 U/L for females), and anti-HCV (Cobas, Roche Diagnostics, Mannheim, Germany) were analyzed in a central laboratory. When anti-HCV was positive, HCV-RNA was determined by PCR (Cobas Amplicor version 2.0, Roche Diagnostics, Mannheim, Germany, lower limit of detection 15 IU/mL).

The study was prospectively performed from September 2019 to October 2020 within a preventive medical examination for adults > 18 years named “Check-Up”. The “Check-Up” is covered by German health care insurance and is usually performed by primary care physicians [14].

The project was approved by the Ethics Committees of the University of Leipzig (ethics vote 338/18-ek) and of the medical association Berlin (ethics vote Eth-X-1/19). All patients provided written informed consent.

All data were analyzed using the software R, v 4.0.4. Only descriptive statistics were used, with the mean (standard deviation) or median [interquartile range] as the chosen summary statistics depending on the distribution of the underlying data.

## 3. Results

**Patient cohort.** A total of 622 patients (mean age 45 years, 44.5% male, 1.3% with a history of injected drug use, 16% immigrants from HCV endemic countries) were recruited by 10 primary care private practices. Five individuals (0.8%) were anti-HCV positive, one was HCV-RNA positive. Patient characteristics are summarized in Table 1.

**Test performance of Xpert HCV Viral Load Fingerstick assay.** The Xpert test was valid in 546/622 (87.8%) patients (Table 2). It was negative in 544 individuals and positive in 2 cases, both of whom were anti-HCV negative. ALT and AST values in these two patients were not indicative of acute hepatitis C infection; therefore, the Xpert test results were interpreted as false positive (false positive rate of 0.4%, 95% CI 0.1 to 1.3%). The HCV-RNA PCR and the Xpert test were both negative in 4/5 anti-HCV positive cases, and the individual with HCV-RNA 4.5 × 10^6^ IU/mL was not detected by the Xpert test.

**The practicability of Xpert HCV Viral Load Fingerstick assay.** Primary care physicians rated the Xpert test practicability as bad, satisfactory, or good in 6%, 13%, and 81%, respectively (Table 3). Sub-analysis of 6 practices performing at least 25 Xpert tests revealed that 14/29 (48%) bad test rankings were judged by one primary care practice, which classified 27% of Xpert tests as bad at their center, whereas the other centers classified the test practicability as bad in only 0–6% of cases and reported good test experience in 76–97% (Table 4).

Test practicability was analyzed in detail with a questionnaire, which was returned by 7/10 private practices (Table 5). The Xpert test could be incorporated into the routine workflow without any difficulties at all centers. Three private practices reported frequent difficulties with the capillary blood draw, and one had frequent difficulty with the use of the minivette. 

## 4. Discussion

In order to eliminate HCV infection, several key factors have been identified in high-income countries, including political will, financing a national program, implementing monitoring of existing programs, screening, awareness, and linkage to care [15]. In Germany, the political will to eradicate the infection is laid out in the BIS2030 strategy [2]. The key factors “awareness and screening at the primary care level” have been addressed recently in a clinical study, which proved the efficacy of guideline-defined screening efforts within a widely used German health care program named “Check-Up” [14]. In response, health care authorities decided to include anti-HCV screening routinely in this “Check-Up” program, which will further strengthen the importance of primary care physicians in the HCV elimination strategy [16]. However, subsequent linkage to care may be impaired, and point-of-care HCV-RNA assays enabling detection of HCV-RNA and diagnosis of active infection in a single visit are important to optimize the care continuum [6,7]. Current European HCV guidelines emphasize the value of point-of-care test systems for widespread population testing, which only need to have a lower limit of detection 1000 IU/mL because the low incidence of a false-negative test for viremia with these assays is outweighed by the benefit of scaling up access to diagnosis and care [17]. The Xpert HCV Viral Load Fingerstick assay fulfills these EASL criteria and therefore deserves further evaluation at the primary care level. According to our data, the test can be successfully included in the daily workflow of primary care physicians; however, the test was only valid in 87.8% of patients. To improve interpretability and practicability in daily routine, the staff could be further trained in the capillary blood draw, or the necessary blood volume (100 µL) should be further optimized. Data from people who inject drugs indicate that the majority of individuals prefer a fingerstick capillary blood draw compared with venipuncture [18]; nevertheless, it should also be acknowledged that blood from venipuncture can be used for further analyses, and that venipuncture is rarely the greatest limitation for patient acceptance [10]. In our study, we unfortunately were not able to assess the patient comfort with the Xpert HCV Viral Load Fingerstick assay.

Whether the use of the Xpert point-of-care test system really improves linkage to care and treatment uptake of antiviral therapy in German real life cannot be judged with our study. In addition, the limited number of participating centers impairs generalizability, and with the cohort size, only imprecise estimates of diagnostic properties of the Xpert HCV viral load test are available. 

However, diagnosis of HCV viremia at the primary care level will be of increasing importance if the “Check-Up” screening strategy is to be successful: In our first screening project carried out in the years 2012/13, the anti-HCV and HCV-RNA prevalence were 0.95% and 0.43%, respectively [14]. In a confirmatory project from the years 2018–2020, the anti-HCV and HCV-RNA prevalence had dropped to 0.5% and 0.1% after therapy with modern direct antiviral agents was widely used [19]. Thus, the gap between positive anti-HCV screening results and viremic patients increased from 50% to 80%. To improve linkage to care of HCV-RNA positive patients, either a reliable direct on-site HCV-RNA test may be beneficial or close collaboration between the primary care physician and the cooperating central laboratory that analyzes the anti-HCV screening test: German health care authorities have adapted the system to include one anti-HCV screening in the “Check-Up” program without budgetary disadvantage for the primary care level [16]. If the anti-HCV screening result is positive, the best strategy is to test automatically for HCV-RNA (reflex RNA testing), which has now been implemented in Germany [16].

In conclusion, our study proves adequate acceptance and practicability of the Xpert HCV Viral Load Fingerstick assay at the primary care level, though its interpretability and diagnostic performance in this setting must be evaluated further before use can be recommended, and the test may become one further piece of the puzzle to eradicate HCV infection in Germany by the year 2030.

## Figures and Tables

**Table 1 viruses-13-02327-t001:** Patient characteristics.

Age, Mean (sd)	45.4 (14.2)
Sex, male/female	277 (44.5%)/345 (55.5%)
ALT in U/L, median [IQR]	29.0 [16.0, 31.8]
above ULN	64 (10.3%)
AST in U/L, median [IQR] (*n* = 1 missing)	26.0 [18.0, 27.0]
above ULN	24 (3.9%)
GGT in U/L, median [IQR]	24.0 [13.0, 27.0]
above ULN	53 (8.5%)
history of injected drug use (*n* = 8 missing)	8 (1.3%)
Immigration from HCV endemic countries (*n* = 8 missing)	102 (16.6%)

**Table 2 viruses-13-02327-t002:** Test performance of Xpert HCV Viral Load Fingerstick assay compared with the Cobas anti-HCV test.

		Xpert HCV			
		Positive	Negative	Invalid	
**Cobas anti-HCV**	Positive	0	5	0	5 (0.8%)
	Negative	2	539	76	617 (99.2%)
		2 (0.3%)	544 (87.5%)	76 (12.2%)	622

**Table 3 viruses-13-02327-t003:** Test performance of Xpert HCV Viral Load Fingerstick assay compared with test rating.

		Result of Test			
		Negative	Invalid	Positive	
**Test rating**	Bad	8	28	0	36 (5.8%)
	Satisfactory	63	15	0	78 (12.5%)
	Good	469	33	2	504 (81.0%)
	No answer	4	0	0	4 (0.6%)
		544 (87.5%)	76 (12.2%)	2 (0.3%)	622

**Table 4 viruses-13-02327-t004:** Practicability of Xpert HCV Viral Load Fingerstick assay.

		Practice (with at Least 25 “Xpert Tests”)					
		F	K	L	M	P	R
**Test rating**	Bad	4 (3%)	0 (0%)	14 (27%)	0 (0%)	2 (6%)	9 (4%)
	Satisfactory	12 (9%)	1 (3%)	2 (4%)	5 (6%)	5 (15%)	50 (20%)
	Good	118 (88%)	36 (97%)	36 (69%)	81 (94%)	26 (79%)	178 (76%)

**Table 5 viruses-13-02327-t005:** Practical evaluation of single steps of Xpert HCV Viral Load Fingerstick assay.

	Never	Rarely	Frequently	Always
Difficulties with capillary blood draw?		4 (57%)	3 (43%)	
Difficulties with the use of the minivette?	5 (71%)	1 (14%)	1 (14%)	
Difficulties in filling the cartridge?	5 (71%)	2 (29%)		
Difficulties with interpretation of the test result?	5 (71%)	2 (29%)		
Was the test result available within the expected 60 min?	1 (14%)		1 (14%)	5 (71%)

## Data Availability

Data can be shared upon request.

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
