# Peer review of "The Practicability of the Xpert HCV Viral Load Fingerstick Point-of-Care Assay in Primary Care Settings"

_viruses, 2021, doi:10.3390/v13112327_

Round 1
Reviewer 1 Report
HCV occurs in all regions. The burden of disease is greatest in the Eastern Mediterranean and Europe Region, with 12 million people chronically infected in each region. Globally, 71 million people are living with the hepatitis C virus (HCV). Despite the availability of effective direct-acting antiviral drugs, only 20% of this population has been diagnosed.
In May 2016, the World Health Assembly adopted the first Global Health Sector Strategy on Viral Hepatitis 2016-2021. The strategy aims to eliminate viral hepatitis as a public health problem by reducing new viral hepatitis infections by 90% and reducing viral hepatitis deaths by 65% by 2030.
Because new HCV infections are usually asymptomatic, few people are diagnosed with a recent infection. In people who develop chronic HCV infection, the infection is often underestimated because it remains asymptomatic until several decades after infection, with symptoms secondary to serious liver damage.
HCV infection is diagnosed in two stages: testing for HCV antibodies using a serological test identifies people who have been infected with the virus. If the HCV antibody test is positive, a PCR test for HCV RNA is needed to confirm chronic infection.
This two-step diagnostic method requires multiple visits to a healthcare practitioner, which results in fewer people being diagnosed with HCV RNA.
For clinical use, on-site HCV RNA assays are important to detect HCV RNA and diagnose active infection in a single visit. The Xpert HCV RNA Viral Fingerstick Assay is designed to improve the diagnosis of viremic infection at the point of care without the need for a follow-up visit. Several early studies (F. M. J. Lamoury et al, 2018; Z. Mohamed et al, 2019; J. Grebely et al, 2020) have demonstrated adequate acceptance and usefulness of this assay at the primary health care level. The authors of the reviewed article also confirmed the applicability of this test for point-of-care diagnostics of HCV, although they recommended further evaluating its interpretability and diagnostic efficacy.
The manuscript can be recommended for publication.
Author Response
We thank reviewer #1 for the evaluation of our manuscript.
Specific comments to be addressed in a point-to-point response were not raised.
Reviewer 2 Report
The authors evaluated the usability of an HCV RNA point of care test (Xpert Fingerstick) in the primary care setting (10 private practices). They found that while rating of this test by primary care physicians varied by center, it was good in 81% of the cases.
Direct RNA testing with this assay was compared to the standard two-step algorithm (antibody test and, if positive, RNA test to confirm viremia). The POC test was assessable in 87.8% (a definition of assessable is missing), positive in two seronegative patients, and negative in one patient that was viremic by the reference method. These results let the authors to conclude that further evaluation in this setting is required.
Major comments:
- The authors should explain why they wished to evaluate direct RNA testing with the GeneXpert instrument in a setting with such low HCV prevalence (it has been recommended in high prevalence settings), and clarify if false positive results were obtained (the false positivity rate is related to the prevalence of the infection in the target population), so that results could be more useful to the reader.
- Please show the results of the Xpert assay in comparison with the traditional diagnostic algorithm in a two by two table. Regarding the two RNA positive cases by the Xpert assay that were negative for antibodies, were they tested for RNA in plasma? This test should be done in order to clarify wether these two cases were false positives of the Xpert assay, or could have been acute infections within the antibody window period. If not performed, did these two cases have elevated transaminases?
Minor comments:
- Methods L61: the time to result of the CE marked Xpert HCV Fingerstick assay is 58 minutes (not 108, as referenced from preliminary studies with a prototype).
- Methods L64: please clarify if the described parameters were tested from venipuncture blood collected in parallel to the capillary blood sample tested for HCV with GeneXpert.
- Results L82: a 16% of migrants is mentioned. Please describe if they were from HCV endemic countries or not. Please consider replacing "a history of iv drug abuse" by "a history of injected drug use".
- Results L86: how was "assessable" defined? why were the rest of cases non-assessable (what could have been improved)?
- Results L104: please replace "minipipette" by "minivette".
- Results/Methods: Were Xpert results delivered to patients? What was the "estimated time interval" considered in Table 4?
- Discussion L149: please include reflex RNA testing (antibodies and RNA are detected from the same blood draw) in the discussion and cite the appropriate literature, as it has already been implemented in many centralized laboratories.
- Discussion: the authors should consider including the study limitations together in one paragraph before the conclusion.
Author Response
We thank reviewer #2 for the comments.
Our answers are included point-to-point in these comments and written in italics.
The authors evaluated the usability of an HCV RNA point of care test (Xpert Fingerstick) in the primary care setting (10 private practices). They found that while rating of this test by primary care physicians varied by center, it was good in 81% of the cases.
Direct RNA testing with this assay was compared to the standard two-step algorithm (antibody test and, if positive, RNA test to confirm viremia). The POC test was assessable in 87.8% (a definition of assessable is missing), positive in two seronegative patients, and negative in one patient that was viremic by the reference method. These results let the authors to conclude that further evaluation in this setting is required.
Major comments:
- The authors should explain why they wished to evaluate direct RNA testing with the GeneXpert instrument in a setting with such low HCV prevalence (it has been recommended in high prevalence settings)
We have now added the following rationale in the introduction:
A WHO report notes that the assay “is intended for use as an aid in the management of HCV infected patients undergoing antiviral therapy” and “is not intended to be used as a blood donor screening test for HCV”. However, since it “quantifies HCV genotypes 1–6 over the range of 10 to 100,000,000 IU/mL” and recent European recommendations state that “screening for HCV infection can be replaced by low-cost point-of-care tests for viraemia with a lower limit of detection 1,000 IU/ml”, it is important to generate evidence for its diagnostic properties in the screening setting.
- and clarify if false positive results were obtained (the false positivity rate is related to the prevalence of the infection in the target population), so that results could be more useful to the reader.
In the results, we have added the information that the false positive rate is 0.4%, 95% CI 0.1 to 1.3%.
- Please show the results of the Xpert assay in comparison with the traditional diagnostic algorithm in a two by two table.
We now provide this table (new Table 2).
- Regarding the two RNA positive cases by the Xpert assay that were negative for antibodies, were they tested for RNA in plasma? This test should be done in order to clarify wether these two cases were false positives of the Xpert assay, or could have been acute infections within the antibody window period. If not performed, did these two cases have elevated transaminases?
The RNA test was only performed if the anti-HCV test was positive (see Methods), so this information is unavailable to us.
We checked the ALT and AST values of the Xpert positive/anti-HCV negative patients: In individual #1, ALT and AST were within normal limits. In individual #2, ALT was elevated < 1.5x ULN and AST was within normal limits. These results are not suspicious of acute HCV infection. Thus, we believe that the two Xpert results were false positive. We added this information in the results.
Minor comments:
- Methods L61: the time to result of the CE marked Xpert HCV Fingerstick assay is 58 minutes (not 108, as referenced from preliminary studies with a prototype).
Thank you, we corrected this.
- Methods L64: please clarify if the described parameters were tested from venipuncture blood collected in parallel to the capillary blood sample tested for HCV with GeneXpert.
It was after venipuncture and we have added this information.
- Results L82: a 16% of migrants is mentioned. Please describe if they were from HCV endemic countries or not.
Good point. We have added this.
- Please consider replacing "a history of iv drug abuse" by "a history of injected drug use".
Okay, we have done so.
- Results L86: how was "assessable" defined? why were the rest of cases non-assessable (what could have been improved)?
We have changed the term to “valid/invalid” for clarity and now define it in the Methods.
- Results L104: please replace "minipipette" by "minivette".
Okay.
- Results/Methods: Were Xpert results delivered to patients? What was the "estimated time interval" considered in Table 4?
The results were not delivered to the patients in this study setting.
We have changed the text in the table to read “Was the test result available within the expected 60 minutes?”
- Discussion L149: please include reflex RNA testing (antibodies and RNA are detected from the same blood draw) in the discussion and cite the appropriate literature, as it has already been implemented in many centralized laboratories.
Quite true. We have added this to the discussion.
Discussion: the authors should consider including the study limitations together in one paragraph before the conclusion.
We have added two further limitations and made them more prominent by “collecting” them in a single paragraph.